# Exploring Migrant Students' Attitudes towards Their Multilingual Identities through Language Portraits

Antonia Stavrakaki and Peggy Manoli *

School of Humanities, Hellenic Open University, 26335 Patra, Greece; std510669@ac.eap.gr
* Correspondence: manoli.polyxeni@ac.eap.gr

**Abstract:** The increasing linguistic and cultural diversity in contemporary societies inevitably affects the field of education by challenging teachers to cope with the coexistence of different languages in the classroom. The present research was intended to investigate migrant children's attitudes towards languages through language portraits in order to help educators obtain insights into student multilingual repertoires and experiences. To this end, by adopting a qualitative approach, the study used linguistic portraits and semi-structured interviews to collect the data. The participants of the study were 10 primary school children whose ages ranged from 8 to 12 with a migrant background who have been living in Greece, particularly on the island of Crete. Using the method of content analysis, the findings of the study indicated that migrant children made specific color choices based on flags, emotions, and world experience, and they put colors on parts of the body according to their functions, which signified students' multilingual identities and experiences. Moreover, the findings highlighted multilingual students' need to negotiate their multiple linguistic repertoires, make choices between the languages, prioritize them, rank them, or give priority to the second language, Greek, without, however, abandoning their first languages. The present study aspires to contribute to the relevant research and draws implications for the implementation of multilingual education and culturally sustaining pedagogies.

**Keywords:** multilingualism; refugee/migrant identities; language portraits; multilingual education; linguistic and cultural diversity





## 1. Introduction

Although migration is one of the oldest events in history, during the last few years. there has been an unprecedented refugee influx mainly because of wars, which brought different groups of people and languages in contact by causing superdiversity in societies and educational settings [1]. Greece, particularly, being at the southeastern tip of the European Union, has been a host country for migrants, refugees, and repatriates from other Balkan and Eastern European countries, as well as countries from the Middle East and Asia, since the 1990s. However, immigration skyrocketed in the last few years because of the refugee crisis in 2015 and the outbreak of the Ukrainian war in 2022. The number of refugee/migrant children with ages from 5 to 17 years in Greece was approximately 45,000 in 2019 ref. [2], while there were 14.630 refugee arrivals from Ukraine in 2022, of which 31.48% were minors [3], which highlighted the need for either starting or continuing school education in the host country by considering the universal right to education for refugees and migrants ref. [4]. In this context, students from different linguistic and cultural backgrounds coexist in classrooms that bring with them multiple languages and identities, as well as change the sociolinguistic realities of classrooms. In order to deal with the linguistic and cultural diversity of classrooms, teachers should be aware of their students' multilingual worlds and, thus, use the students' funds of knowledge ref. [5] and apply multilingual education refs. [6,7], as well as culturally sustaining pedagogies ref. [8]. When a student's funds of knowledge are valued within a classroom, they feel empowered to

build their new lives, acquire school knowledge, and bridge the gap between the languages and cultures they speak or experience ref. [7].

There has been an increasing interest in the field of sociolinguistics in exploring individuals' multilingual identities through different methodologies, especially in school contexts refs. [9–15]; a new type of visual representation of student linguistic repertoires that has drawn researchers' interests has been the so-called language portraits refs. [9,16]. More specifically, language portraits are body silhouettes which make student linguistic repertoires visible through the use of colors that reveal multilingual student experiences, practices, and attitudes towards languages ref. [9]. In fact, colors are considered signifiers and serve as meaning potentials in a given socio-cultural context ref. [17]. Researchers draw a distinction between two different sources for making meaning: association and distinctive features. While distinctive features of colors demonstrate values on a range of scales, such as saturation and darkness, association refers to the source that the particular color derives from, which can be either a certain object or a place, thus indicating a symbolic and emotive value in a given socio-cultural context ref. [17]. In this way, language portraits are seen as alternative ways of expressing multilingual student emotions, practices, and experiences that are linked to each language ref. [16]. In particular, ref. [14] explored twenty-six elementary students' cultural and linguistic self-portraits, along with their interpretive narratives, in Canada, and they revealed that the students' self-portraits depicted their cultural and linguistic skills through multiple layers of color and symbols, which highlighted that students connect various languages and cultures with different roles and functions in their lives. Simultaneously, ref. [10] investigated seven young multilingual learners' identities in Canada through the use of language portraits and found that student linguistic identities were expressed in terms of expertise, affiliation, and inheritance through flags, as well as the symbolic placement of their first languages (L1s) on the body of their language portraits. Ref. [18] investigated the impact of using language portraits on improving teachers' understanding about their student languages and identities and revealed thematic patterns in relation to identity, culture, and language use that highlighted that their L1s were put on specific body parts of the portraits to indicate intimacy and close connection, while a less emotional connection was found to the second language (L2), which was associated with instrumental uses, such as future academic studying or better job prospects. Refs. [12,13] worked with visual narratives with Portuguese migrant children in Germany, and the visual narratives proved to be a useful tool to gain access to the children's languages, patterns of experience, and feelings, as well as represent how multilingual minds make sense of the relationship between their linguistic repertoires. Ref. [14] used 570 language portraits and 21 semi-structured interviews to make young students' multilingual repertoires visible, wherein they revealed that students were aware of their multiple linguistic repertoires; the findings of the study also indicated that, in terms of color choices, there were associations with national flags, their own previous experiences, and personal feelings, while, in terms of language placement, it was shown that languages were structured according to linguistic skills and body parts. Moreover, one of the aims of the study conducted by ref. [19] was to explore young migrants' linguistic repertoires, and they revealed that they tended to locate their L1s on the central parts of the silhouette, such as the heart or the head, and that the relation between languages and nations through the use of flags was prominent; the findings also indicated that these children had diverse linguistic repertoires, which they tended to separate according to the linguistic domains of school and home, which highlighted the dichotomy between their multilingual realities and the monolingualism in schools, even though often the drawing lines often became hybrid and fluent. In the Greek context, particularly, which the present study focuses on, there was scarce research on mapping multilingual young students' sociolinguistic profiles through language portraits. For instance, ref. [20] explored both Albanian migrant mothers' and their children's attitudes to multilingualism through narratives and language portraits that revealed that children's specific color choices were based on flags and world experience, while color placement on parts of the body depended on their functions (e.g., in their hands

when they often use the language), thus highlighting the children's overall perception of multilingualism as a qualification or skill ref. [12]. Another study ref. [21] investigated an unaccompanied minor's linguistic repertoires and ideologies by using a language portrait that indicted that specific parts of the body were chosen for specific languages; for example, the L1 was placed on his face, while the L2 was placed on lower parts of the body.

Considering, on the one hand, the importance of language biographies, especially the use of creative tools, such as language portraits with young children, for exploring multilingualism in the field of sociolinguistics in school contexts ref. [16] and, on the other hand, the rather limited research in the field, particularly in Greece, the present study aimed to explore 10 migrant children's attitudes towards their multilingualism through language portraits and semi-structured interviews. Simultaneously, according to the research, Greece adopts a monolingual policy and, thus, refugee/migrant student's linguistic and cultural funds remain 'invisible' in Greek state schools, as teachers mainly focus on the development of their L2 skills ref. [22]. In this way, the study aspires to contribute to relevant research and provide insights both for teachers and policymakers into student multilingual realities and practices in order to help them draw on the students' funds of knowledge ref. [5], apply multilingual education refs. [6,7], practice culturally sustaining pedagogies ref. [8], and build a more inclusive educational system and society. In this context, the present study aimed to answer the following research questions:

How do multilingual children represent their linguistic repertoires in language portraits?
What are their overall feelings towards their multilingualism?

## 2. Materials and Methods

The research was conducted in the context of the MA program "Language Education for Refugees and Migrants" of the Hellenic Open University. Considering the research aims and questions, the present study adopted a qualitative approach to the data collection. Namely, semi-structured interviews and linguistic portraits were used in order to delineate the participants' linguistic repertoires and identities. The data collection process began with the researcher's contact with the students' parents, who gave their consent to their children to participate in the research after being informed about the aim of the study. Then, the time and place of the interviews were set. In order to comply with the rules of research ethics, the participants' anonymity was fully ensured, and their personal data and names were not mentioned.

Language portraits, one form of graphic visualization, consist of the outline of a body silhouette, which participants have to color to represent their linguistic repertoires symbolically. As visual representations, they allow participants to present their linguistic repertoires through other semiotic modes, such as the visual or through color to communicate meaning in a given socio-cultural context ref. [17]. The use of language portraits as a research method was popularized by Busch ref. [23], who used them in multilingual schools in Austria and South Africa. In fact, they turned out to be powerful tools that represent multilingual people's lived language experiences, practices, and trajectories, as well as provide a way for teachers to access and understand their students' multilingual realities without restricting them to focus on a specific language refs. [7,14,16]. The silhouette usually creates body-related metaphors, such as "x language is in my heart", thus indicating certain feelings and functions; for example, the L1s are usually put on central body parts that express feelings of intimacy, while languages put on peripheral body parts demonstrate fewer identifications ref. [19]. However, most researchers working with language portraits recognize the limitations of this research tool and, therefore, they use a multi-modal approach to data collection by either interviewing participants or asking them for a written/oral description of their language repertoire refs. [12,13,16]. According to ref. [23], when visual representations are accompanied by interviews or narratives, they give researchers an overview of how participants conceive of and depict their linguistic repertoires and experiences. In this way, language portraits cannot be analyzed independently of each speaker's descriptions and explanations, but are used as a starting point

for interviews or narratives. For this reason, we also used a multi-modal approach to our research, that is, language portraits were accompanied by the participants' interviews. Namely, after students having finished coloring the silhouettes the way they wanted while thinking about their languages, the interviews were conducted. In terms of the language portraits, it should be clarified that the colors used to depict the languages spoken were the student's choice.

The interview is the most common method used in social science research to extract in-depth information, especially when the interest of the research focuses on the subjective meaning of exploring personal or social processes, experiences, or attitudes, which offers people the opportunity to be heard and share their views, experiences, and life stories [23]. In particular, semi-structured interviews were used, including pre-determined open-ended questions, which provide researchers the opportunity to constantly monitor the direction and depth of the interview and, at the same time, allow interviewees to answer questions in a more flexible way. Thus, 10 semi-structured interviews were conducted in Greek and lasted from 24 to 37 min, with a focus on the main research questions. The children's interviews were recorded and then transcribed.

The participants of the study were all primary school children with ages ranging from 8 to 12 and a migrant background who have been living in the prefecture of Eastern Heraklion, in the Municipality of Hersonissos, on the island of Crete, Greece. The sample of the study could be regarded as a convenience sample [23], because one of the researchers came from the island of Crete and was working at this institution. Six of them were boys and four of them were girls. Their countries of origin varied: most of them came from Albania and Bulgaria, two of them came from Russia, one came from Canada, and one came from Latvia (see Table 1 below). Most of them were second generation migrants, whereas three of them had been living on the island for almost 4 years. Furthermore, most of the participants had some shared languages, such as Greek used as an L2 and English as a foreign language, while some shared an L1, such as Albanian or Russian, as they came from the same country, or they were siblings. However, one child, who has been in Heraklion for only a year, did not know Greek very well and did not fully understand some questions. In fact, one of his friends helped him with the interview process by resorting to translation where necessary. It should be mentioned that the information on which languages the students could speak was obtained through direct questions posed at the beginning of the interviews.

**Table 1.** Participants' profiles.

| | | | | | Languages Spoken | | |
|---|---|---|---|---|---|---|---|
| Participants | Age | Gender | Years in Greece | Country of Origin | L1 | L2 | Foreign Languages |
| P1.M | 8 | girl | 7 | Turkey | Turkish–Bulgarian | Greek | English |
| P2.A | 8 | girl | 4 | Latvia (mother)—Greece (father) | Russian | Greek | English |
| P3.E | 10 | boy | 10 | Albania | Albanian | Greek | English–German |
| P4.A | 10 | boy | 10 | Albania | Albanian | Greek | English–German |
| P5.O | 12 | girl | 12 | Albania | Albanian | Greek | English |
| P6.1 | 11 | boy | 11 | Albania | Albanian | Greek | English |
| P7.M | 11 | girl | 5 | Russia | Russian | Greek | English |
| P8.M | 13 | boy | 5 | Russia | Russian | Greek | English |
| P9.A | 12 | boy | 1 | Russia | Russian | Greek | English |
| P10.L | 12 | boy | 12 | Canada | English | Greek | French |

The interview data were analyzed through the content analysis method focusing more on qualitative aspects of the content, i.e., the absence/presence of specific elements in the content, rather than the frequency in which they appear ref. [3], in order to formulate a judgment as to what the content means. Namely, the researchers independently studied the data after interview transcriptions and grouped relevant meanings into categories and subcategories according to the research aims and questions by attributing different codes to each category to organize and reduce the data.

## 3. Results

### 3.1. Participants' Visual Representations of Their Linguistic Repertoires

This section presents the findings of the study coming both from the language portraits and the participants' interviews in order to make the participants' linguistic repertoires visible. An attempt was made to identify the most frequent patterns of color choices and language placements in their language portraits. In fact, three recurring patterns were revealed in terms of their color choices, and two patterns were revealed regarding their language placement on the portraits, which are thoroughly presented below.

3.1.1. Participants' Color Choices

Flags

A frequent pattern observed within the sample (5 out of the 10 language portraits) was that color choices were made according to the colors present in the national flag (e.g., choosing red, as it is the main color in the flag of Albania, as well as in the Canadian flag). The above finding was verified by the interview process:

**I:** Why did you choose red to represent Albanian? P3.E: Red is on the Albanian flag.

**P4.A:** Why on the flag that Albania has... it is red. P8.M: Because the Russian flag is red...I like red...as I do for the Russian language. P10.L: As you can see . . . . I tried to draw the flag of Canada.

As seen in the excerpts, the participants justified the colors chosen for their language portrait in association with the colors being present in the national flags, thus making a direct connection between nations and languages.

In Figure 1 below, we can see that one of the participants chose to draw the whole Canadian flag in order to represent a language instead of picking just one of the colors of the flag.

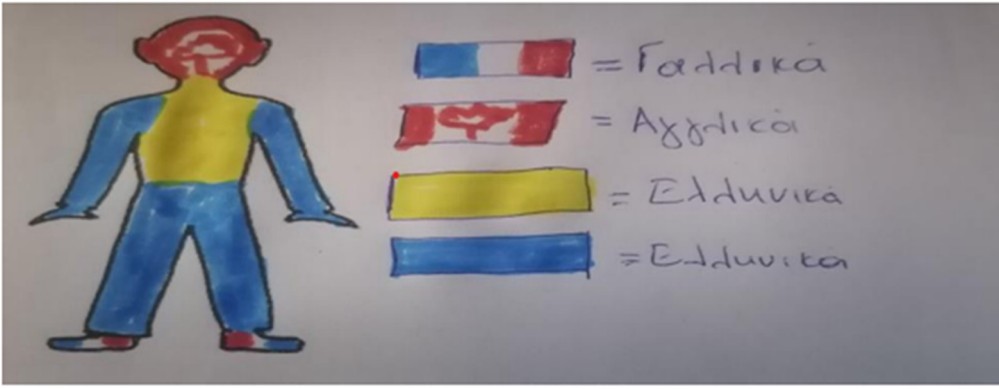

**Figure 1.** P10.L: Language portrait using color choices according to the national flags.

World Knowledge and Personal Experience

Some of the participants' color choices were not as straightforward as the one mentioned above. As has been observed, 3 out of the 10 participants associated colors with their own world knowledge; that is, they tended to choose colors that reminded them of

the experiences they lived. Therefore, these choices varied from participant to participant. In the example below, two of the participants chose the yellow color to represent Greek based on their own perceptions of the country:

**I:** Why did you choose yellow to represent Greek?

**P7.M:** Because Greece has a lot of sun and that's why I chose yellow, which is like the sun.

**P9.A:** Because Greece has a lot of sun.

**I:** Why did you choose blue to represent Greek?

**P2.A:** Blue is the sea of Greece.

Simultaneously, participant 9 used his everyday experience in Russia to justify the use of the green color to represent the Russian language (as it is also evident in Figure 2 below).

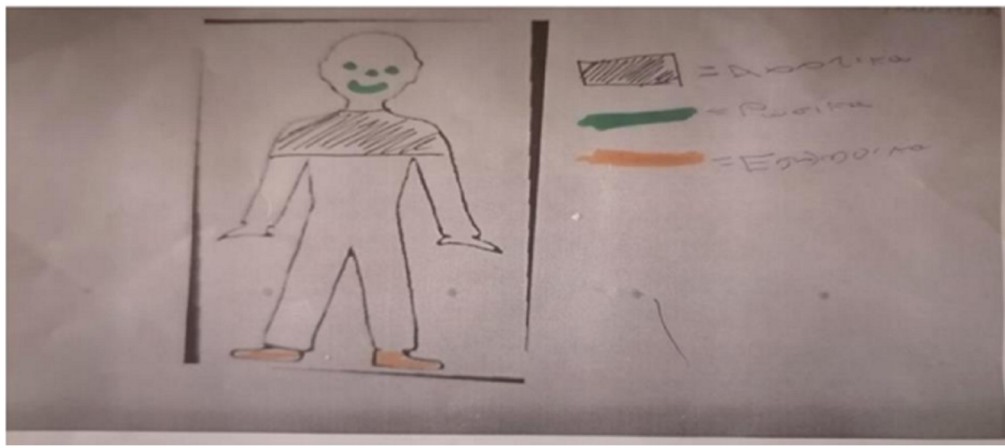

**Figure 2.** P9.A: Language portrait using world knowledge and personal experiences.

**I:** Why did you choose green to represent Russian?

**P9.A:** Because Russia has forests.

Similarly, orange was chosen to represent Russian because of an experience a participant had when she used to live in Russia (see also Figure 3 below):

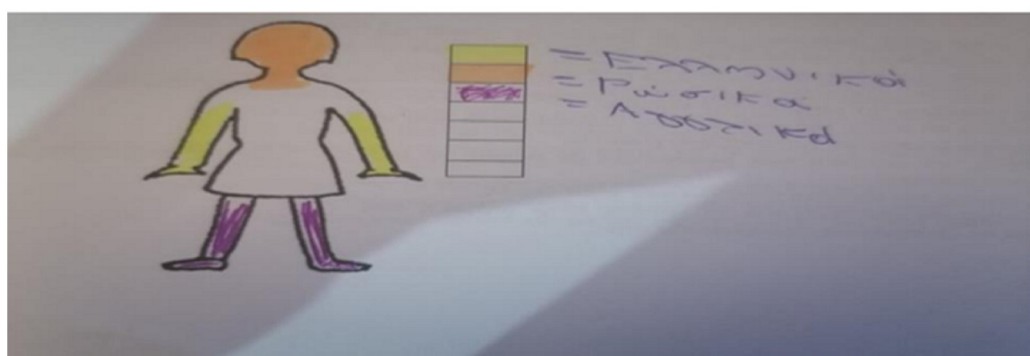

**Figure 3.** P7.M: Language portrait using world knowledge and own experiences.

**I:** Why did you choose orange to represent Russian?

**P7.M:** Oh, now it's a bit hard to explain, because in Russia at Christmas every

Christmas Russians eat tangerines. I don't know where they get them, but their color is orange.

In this excerpt, the participant's positive experience on Christmas holidays determined the choice of the color used to represent the Russian language.

Emotions

Another pattern used by 3 of the 10 participants was that the color choice was based on emotions that the participants had towards these colors; namely, they tended to relate their favorite colors to the language to which they were feeling closer. Figure 4 is an indicative representation of the emotional significance that colors can carry:

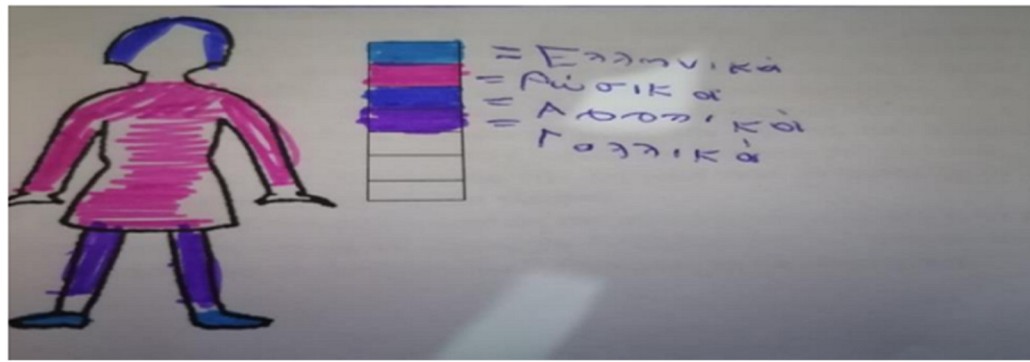

**Figure 4.** P2.A: Language portrait drawing on connections between favorite color and favorite language.

**I:** Why did you choose pink to represent Russian?

**P2.A:** Pink is my favorite color, and so is the Russian language.

**P4.A:** Red for Albanian and my favorite color.

**P8.M:** Because the Russian flag is red...I like red...as I do for the Russian language (see also Figure 5 below).

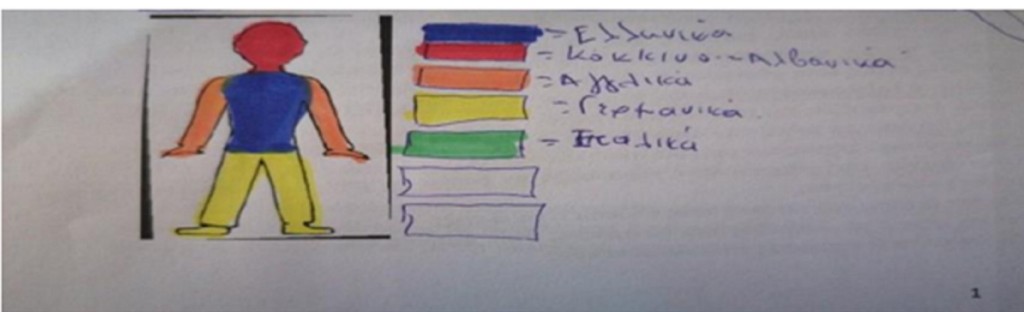

**Figure 5.** P4.A: Language portrait drawing on connections between favorite color and favorite language.

In the above examples, participants clearly established a positive connection between emotions/preferences and the colors used in their language portraits.

3.1.2. Participants' Language Placements
Attachment

When it comes to the placement of the colors on the silhouette, a frequent pattern found was that of placing the languages on silhouettes in accordance to the participants' perceived attachment to a language. Namely, they would place the languages they felt closer to on the upper or central parts of the body, and they placed the languages they felt not so close to on the lower or peripheral parts (see Figure 6 below). Almost all of the interviewees chose this pattern for their language placement on the portraits, as it became clear from the excerpts below:

**I:** Why did you choose red to represent Albanian?

**P3.E:** Red is on the Albanian flag.

**I:** What does it mean that you painted Albanian on your little head?

**P3.E:** Although I grew up in Greece, I think like an Albanian....to keep them in mind (see also Figure 6 below).

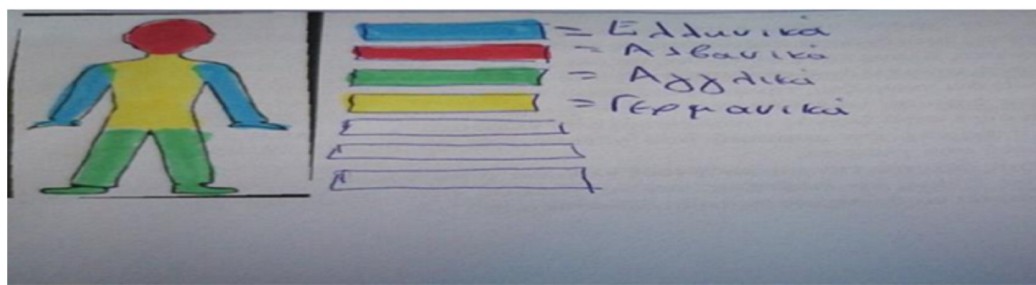

**Figure 6.** P3.E Language portrait concerning language placement according to the perceived attachment to a language.

P3.E made a clear connection between the attachment to his L1 and its placement onto his head. As it became evident from his interview that accompanied the analysis of his language portrait, his choice to put the color of the Albanian flag onto his head highlighted his strong attachment to it, which, according to his words, helped him think as the Albanian people did. Likewise, P2.A chose to put her favorite color onto her main body as well (where the heart is), thus highlighting her strong attachment to the specific language. For example:

**I:** Why did you choose pink to represent Russian?

**P2.A:** Pink is my favorite color.

**I:** What feelings arise when you think about your L1? Is your L1 important for you?

**P2.A:** I love Russian; for me, it's mom and dad.

Body Functions

Another frequent pattern indicated by the research findings was that of placing their languages on the body silhouettes according to the functions of each body part, such as, for instance, placing languages they used at school onto their hands or placing languages they used to speak on their mouth. The majority of the participants followed the specific pattern while placing their languages on the portraits. According to the participants' words below, we can see that the participants related their languages to body parts by creating common metaphors:

**I:** What does painting Greek on your little hands mean?

**P3.E:** I do things with my hands in Greek and I write at school.

**P5.O:** Greek helps me get along with the kids at school.

**P7.M:** Because all the lessons we do at school are in Greek, and I do all this with my hands that is, I write with my hands... that's why I chose hands.

**I:** What does painting Greek on your little feet mean?

**P8.M:** Because I know Greek less but also because I also walk in Greece.

**I:** What does painting Russian on your mouth mean?

**P9.A:** I can speak Russian better than Greek.

Overall, most participants justified the color choice and language placement according to the functions and identifications they shared with them. For example, red in their hearts symbolized the importance of the L1 in their lives (L1s are usually put on central body parts), while placing a language onto their head pointed to the importance of this language for their social inclusion and future choices. Some participants frequently added colors to

the extremities of the silhouettes (head, arms, and feet) and central parts (heart and belly), as these were the body parts they could relate functions to (e.g., understanding, writing, walking) or common metaphors to the languages they wanted to represent.

*3.2. Students' Perceptions of Multilingualism and Emotions*

In order to answer the second research question referring to students' perceptions of multilingualism and emotions, both the data coming from the interviews and those from the language portraits were considered. In other words, the findings indicated that almost all children developed positive emotions towards their multilingualism thus highlighting, simultaneously, their need to use the various languages according to domains, interlocutors, and communicative situations (e.g., P2.A, P4.A, P5.O, P7.M, and P10.L).

More specifically, according to a student P2.A's words: "I love both Greek and Russian for me it's mom and dad". Additionally, it was found that all students' emotions towards their L1s were stronger than those towards the Greek language. For example:

**I:** What feelings arouse when you think about your L1?

**P6.I:** "I feel love; I feel my cousins and uncles who also speak Albanian".

**I:** Is your L1 important for you?

**P1.M:** A lot. With mom and dad, we speak Bulgarian so that I don't forget them. When I go to Bulgaria, I want to be able to speak.

With regard to the color choices made to represent their L1s, the choices varied, since P1.M and P2.A associated their favorite colors with their L1s ("I: Why did you choose pink to represent Russian?" "P2.A: Pink is my favorite color and so is the Russian language."), while P3.E, P4.A, P5.O, P6.I, P8.M, and P10.L chose the color according to the national flag, thus indicating a direct association between nation and language. For instance: "I: Why did you choose red to represent Albanian?" "P3.E: Red is on the Albanian flag". With regard to the color choice associated with the Greek language, almost all students chose the blue color, as it was associated with the national flag and the sea of Greece, thus expressing, in this way, their experience on the island of Crete. However, a less strong attachment was attributed to Greek as an L2, as most students embodied the color onto their hands or feet to indicate, for instance, the body functions of using the language at school or walking on the island, respectively, and thus pointing to its instrumental uses, such as, social inclusion, school attendance, and communication. For instance: "P1.M: For me Greek is important so I pay attention to my teacher at school".

## 4. Discussion

As a result of the continuous migration and population movement, students from different linguistic and cultural backgrounds coexist in classrooms and bring with them multiple languages and identities. In order to deal with the linguistic and cultural diversity of classrooms, teachers should be aware of students' multilingual worlds and the reality they themselves experience, because valuing migrant student linguistic and cultural repertoires in classrooms contributes to their emotional, social, and educational development ref. [10]. In this context, the present study aimed at exploring the way migrant children coped with their multilingualism. For the purposes of the study, semi-structured interviews and language portraits were used to collect and triangulate data from ten primary students with a migrant background residing on the island of Crete, Greece. Language portraits, a visual representation of student linguistic repertoires refs. [16,23], were used in socio-linguistic research to explore student multilingualism, which proved to be a powerful tool, both for teachers and students in the educational process refs. [10,12–14,18,19].

More specifically, the findings of the study indicated that most students made specific color choices based on flags, world experiences, and emotions, as well as color placements on the body parts of the portraits depending on body functions and identifications to make their linguistic repertoires visible; they simultaneously unraveled skill development in languages and positive feelings towards their multilingualism, which were further

explained and supported by their interviews. In other words, the data analysis indicated that all students' emotions towards their L1s were stronger than those towards their L2, since they represented these experiences through language placement onto central body parts of the silhouettes, such as their hearts, according to body functions that semiotically underlie, in this way, how intimately they felt towards their L1s, which was also consistent with relevant studies refs. [10,14,18–21]. Additionally, in terms of the Greek language, the findings revealed less emotional attachment but overall positive feelings for speaking the L2 by associating its use with instrumental uses, such as job prospects, school attendance, and social inclusion, since all of them were enrolled in a Greek primary school. That is why, according to the functions of the body, the participants of the study placed the L2 either on peripheral body parts, such as their hands or feet, thereby metaphorically underlying their need to know and use Greek (for example, to be able to write and communicate at school), which was consistent with previous studies refs. [10,14,18–21]. Additionally, the findings of the study supported the meaning potential of color use, particularly regarding association, which refers to the source that the particular color derives from, which can be either a certain object or a place, thus indicating a symbolic and emotive value in a given socio-cultural context ref. [17]. For instance, according to some students' answers, it was found that the yellow color was used in the hands of the language portrait to represent the L2, because it was associated with the sun, which is a feature of Greece, thus indicating the source and the interpretation of its use based on their perception of the country itself (the same was observed for the use of the blue color and associating it with the natural element of sea in Greece). Similarly, a Russian girl chose to color her head orange to represent her L1, which was derived from her world experience related to eating tangerines back in her country on Christmas Eve. Moreover, the color choice in some students' language portraits was based on the colors of national flags, which revealed a strong connection between nation and language that expressed both a symbolic and an emotive value, which concurred with relevant research refs. [10,14,19,20].

To conclude, the findings of the study revealed that children's specific color choices were mainly based on flags, emotions, and world experience, while language placements on body parts of the portraits depended on their functions (e.g., on their hands when they often used the language) and identifications (e.g., on their heart to indicate love and affection). All these visual options signified students' diverse identities, experiences, and practices that highlighted, at the same time, their need to negotiate their multiple linguistic repertoires by making choices between the languages, prioritizing them, ranking them, or giving priority to Greek as an L2 without, however, abandoning their L1. Consequently, it can be concluded that, though children seemed to perceive multilingualism as a qualification or skill ref [19], they treated their linguistic repertoires as parallel monolingualisms to meet the monolingual demands of the Greek school and society ref. [22] as well as, simultaneously, maintain their L1s, which is supported by pertinent studies refs. [12,13,19].

Overall, the study aspires to contribute to relevant research, which is rather limited, and provide insights both for educators into transforming the invisibility of student multilingualism into an obvious advantage for the whole class, especially in Greek state schools where student multilingualism remains invisible ref. [22], and policy makers to include multilingual education refs. [6,7] and culturally sustaining pedagogies ref. [8]. However, considering some of the constraints of the study, such as the rather limited sample, more research is needed, including a larger sample, to generalize and verify the findings of the study. Simultaneously, future studies should explore possible differences in color choice and location on body parts in relation to extra-linguistic (social) variables, such as age, gender, years in Greece, and country of origin, as well as differences depending on whether the languages in each person's repertoire belonged to the L1 or L2. Additionally, quantitative data would give interesting results in relation to body parts, which could be crossed with the conceptualization of body parts in the languages involved.

**Author Contributions:** Conceptualization, A.S. and P.M.; methodology, P.M.; software, A.S.; validation, A.S. and P.M.; formal analysis, A.S.; investigation, A.S.; resources, A.S.; data curation, A.S.; writing—original draft preparation, A.S.; writing—review and editing, P.M.; supervision, P.M.; project administration, P.M. All authors have read and agreed to the published version of the manuscript.

**Funding:** This research received no external funding.

**Institutional Review Board Statement:** The study was conducted in accordance with the Declaration of Helsinki, and approved by the Committee of Research and Academic Ethics of Hellenic Open University (protocol code 55804 and date of approval 3 September 2022).

**Informed Consent Statement:** Informed consent was obtained from all subjects involved in the study.

**Data Availability Statement:** Not applicable.

**Conflicts of Interest:** The authors declare no conflict of interest.

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
