# Peer review of "Exploring Migrant Students’ Attitudes towards Their Multilingual Identities through Language Portraits"

_societies, doi:10.3390/soc13070153_

Round 1

Reviewer 1 Report

The evaluated article is entitled "Exploring migrant students' attitudes towards their multilingual identities through language portraits” and aims to investigate the linguistic attitudes of migrant children in Greece, with the objective of making multilingualism visible in primary education. The results have been obtained through language portraits and semi-structured interviews with ten children. I think it is interesting, as the author indicates, the use of semi-structured interviews to complement the limitations of the portraits. In this sense, the methodology used is well justified. 

I just want to point out a few questions, the answers to which could be added at some point in the article:

-        At no point does the author explain how he/she obtains the information on which languages the students speak. It has been detected that sometimes, if the migrants have a minoritized first language, they do not name it as their first response, and it is necessary to go deeper in obtaining this information. In the list of languages spoken by the informants, there is no presence of minoritized languages and, as we said, we know that the speakers tend to hide them and the majority languages in the territory of origin are overrepresented. How was the information on the languages spoken obtained? Through a direct question? 

-        Who decides which colors are available to the children? Which colors are available to them?

-        Throughout the presentation of the results, references are made to an undetermined number of participants. I think it would be appropriate to give in each case the exact number, since the sample is small.

I think the article is well structured and points out interesting ideas, but should be expanded in further studies. The conclusions could point out limitations and future lines of research. I suggest some issues that I think could be taken into account in a study of this type:

-        Are differences observed depending on whether the languages in each person's repertoire are minority/minoritized or majority/dominant in the territory of origin? I think it would be interesting to see the use of colors and, above all, the location on body parts according to this variable (e.g. Tatar-Russian / Ukrainian-Russian / Berber-Arabic / Punjabi-Udu, etc.) in addition to the host languages.

-        As for the use of colors: what about languages that have no possible association with a flag?

-        It would be worthwhile to analyze the localization of languages to body parts taking into account the conceptualization of body parts in each speaker's L1 (e.g., are there differences according to whether the language shows a different conceptualization of cognitive capacities in the heart or the head, compared to other languages that do not show this dichotomy? Are there differences according to whether the seat of feelings is related to one organ or another (gallbladder, heart, liver, etc.)?

-        I think it would be interesting, in future studies, to enlarge the sample, at least as far as portraits are concerned. Quantitative data would give interesting results in relation to body parts, which could be crossed with the conceptualization of body parts in the languages involved. 

-        If migratory contexts are analyzed, a control group with Greek L1 (in this case) could be included.

Reviewer 2 Report

Dear authors, the topic is interesting. How is it practical value? What new is in the theory of your research?

Line 7 Please avoid or explain vague language:

in order to make young children’s multilingualism visible and help educators get in- 7 sights into student multilingual world.

Please increase the number of participants for your research. Please refer to the standards and recommendations for research participants in pedagogy topics.

Line 17 The present study aspires to contribute to the relevant research and draws implications for the implementation of multilingual education and culturally sustaining pedagogies.

Please avoid vague language .

Please find the theory to sustain discussion of colours rendered in objects and people as now it looks like an interesting association rather than research findings as they adhere to certain criteria.

In section 3.2: please consider turning open ended questions like what feelings arouse when speaking a language into choice offered questions and an option to add ones own information. As the example of the answer given does not provide the information to the question discussed here: line 306; so it is not possible to draw the desired conclusions.

In Section Discussion: please discuss the data obtained; please consider putting the introductory statement into other sections:

Iines 332-333: The continuous migration and population movement brought different groups of people and languages in contact causing superdiversity in societies and educational settings ref.[1]. Please explain the choice of location and institution.

Lines 359 and further: Please consider you need to present your own conclusion instead of already published research conclusions.

 Please refocus your research goal.

English grammar is fair. Change from vague language in science writing

Reviewer 3 Report

Please, find the recommendations for the authors in the PDF upload.

Round 2

Reviewer 2 Report

Dear Authors,

Thank you for the comments and alterations provided. Unfortunately, the approach to planning and conducting research are not met.

Please explain how in the current version line 7 linguistic repertoire effects the color choices and associations made with nationalities.

Lines 85-86 now speak about Albanian mothers and their children attitudes to multilingualism

Then, lines 98-99 now speak about : This language 98 (Greek) has been less studied in the relevant literature with an emphasis on individuals’ 99 multilingual identities through visual representations.

Please refer to the questions asked in my first review: what is the goal you pursur; what gap you would like to overcome and if the results obtain satisfy you; how many results do you need to consider the research has been completed. What languages and nationalities you choose and what are the reasons for this.

Please profound revise your vision for research and review the criteria for research.

At present you may consider publishing a conference paper based on your observations which may be considered as a starting point for serious investigations.

no comments

Round 3

Reviewer 2 Report

Dear Authors,

comments from review 2 remain.

Dear Authors,

You have submitted the identical text with your second submission including responses to reviewers.

It is strongly recommended to give this study research criteria and objectives. See the review 2.

Author Response

Dear Reviewer,

We appreciate your effort and precious time to review our manuscript again and provide insightful comments. Thanks to all the reviewers’ valuable and insightful comments the current version of the manuscript was improved. The authors have carefully considered your comments and did their best to address each of them or further explain points that were not clear to further improve the quality of the manuscript. After careful revision, we hope that the manuscript meets your high standards in its current version.

Below, in the effort to address the reviewer’s comments we provide a thorough description of the rationale of the research design, aims and criteria. All modifications and revisions in the manuscript were marked up using the “Track Changes” function. 

Yours Sincerely,

Response to Reviewer 2

Upon your comments, the content of the manuscript is succinctly described and contextualized with respect to previous and present theoretical background and empirical pertinent research (see section 1. Introduction and section 4. Discussion), where the relevant theoretical framework and basic empirical research is mentioned and discussed. In fact, upon your suggestion, one more pertinent work has been added: “Moreover, one of the aims of the study conducted by ref. [21] was to explore young migrants’ linguistic repertoires and revealed that they tended to locate their L1 on central parts of the silhouette, such as the heart of head, and that the relation between languages and nations through the use of flags was prominent; the findings also indicated that these children had diverse linguistic repertoires, which they tended to separate according to the linguistic domains of school and home highlighting the dichotomy between multilingual reality and monolingualism in schools, though often the drawing lines have become hybrid and fluent” (lines 83-90). Moreover, the research design, aims, questions, and methods comply with the overall research criteria and are clearly stated throughout the manuscript but see, particularly, section 2. Methods and materials. Initially, the aim of the present study was to: “The present research intended to investigate migrant children’s attitudes towards languages through language portraits in order to help educators get insights into student multilingual repertoires and experiences”, which was clearly stated from the very beginning, that is, in the abstract (lines 6-8), in the introduction (lines 102-107), in methods and materials (lines 121-124) and in the discussion (lines 366-368). Regarding the gap the specific study intends to fill in, it is the following: “Considering, on the one hand, the importance of language biographies, especially the use of creative tools, such as language portraits with young children, for exploring multilingualism in the field of sociolinguistics in school contexts ref. [5] and, on the other hand, the rather limited research in the field, particularly, in Greece, the present study aimed to explore 10 migrant children’s attitudes towards their multilingualism through language portraits and semi-structured interviews. Simultaneously, according to research, Greece adopts a monolingual policy and, thus, refugee/migrant student’s linguistic and cultural funds remain ‘invisible’ in Greek state schools, as teachers mainly focus on the development of their L2 skills ref. [11]. In this way, the study aspires to contribute to relevant research and provide insights both for teachers and policymakers into student multilingual realities and practices in order to help them draw on student’s funds of knowledge ref. [12], apply multilingual education ref. [7,8] and culturally sustaining pedagogies ref. [1] and build more inclusive education and society”, which was clearly mentioned in lines 102-114 (see section 1. Introduction, 3rd paragraph) and lines 417-421 (see section 4. Discussion, 4th paragraph): “Overall, the study aspires to contribute to relevant research, which is rather limited, and provide insights both for educators into transforming the invisibility of student multilingualism into an obvious advantage for the whole class, especially in the Greek state schools where student multilingualism remains invisible ref. [11], and policy makers to include multilingual education ref. [7,8] and culturally sustaining pedagogies ref. [2]”. Simultaneously, in order to comply with research criteria at the end of the introduction section the research questions are included: “In this context, the present study aimed to answer the following research questions: How do multilingual children represent their linguistic repertoires in language por-traits? What are their overall feelings towards their multilingualism?” (lines 114-118). In terms of the results, they are clearly presented according to the research aims and questions, which are accompanied by examples of the participants’ exact words or language portraits to further document them (see section 3. Results). Additionally, the basic research findings are discussed in the relevant section making, simultaneously, associations with pertinent studies (see section 4. Discussion).

Taking all the above into account, we firmly believe that the specific study complies with the research criteria, as there are clearly stated research aims and questions, all the methodological choices are thoroughly explained and justified (see section 2. Materials and Methods), the findings are presented according to research aims and questions (see section 3. Results) and discussed according to relevant studies highlighting the contribution of our research to the relevant field (see section 4. Discussion).

Last but not least, according to your insightful comment, the manuscript underwent careful editing to further boost its quality.